# Secondary Cardiovascular Prevention after Acute Coronary Syndrome: Emerging Risk Factors and Novel Therapeutic Targets

**DOI:** 10.3390/jcm12062161

**Published:** 2023-03-10

**Authors:** Angelo Silverio, Francesco Paolo Cancro, Luca Esposito, Michele Bellino, Debora D’Elia, Monica Verdoia, Maria Giovanna Vassallo, Michele Ciccarelli, Carmine Vecchione, Gennaro Galasso, Giuseppe De Luca

**Affiliations:** 1Department of Medicine, Surgery and Dentistry, University of Salerno, 84084 Baronissi, Italy; 2Division of Cardiology Ospedale degli Infermi, ASL, 13875 Biella, Italy; 3Department of Clinical and Experimental Medicine, Division of Cardiology, AOU Policlinico G. Martino, University of Messina, 98122 Messina, Italy; 4IRCCS Hospital Galeazzi- Sant’Ambrogio, 20161 Milano, Italy

**Keywords:** acute coronary syndrome, secondary prevention, coronary artery disease, lipoprotein(a), inflammation, inflammasome, microbiota, emerging therapies

## Abstract

The control of cardiovascular risk factors, the promotion of a healthy lifestyle, and antithrombotic therapy are the cornerstones of secondary prevention after acute coronary syndrome (ACS). However, many patients have recurrent ischemic events despite the optimal control of traditional modifiable risk factors and the use of tailored pharmacological therapy, including new-generation antiplatelet and lipid-lowering agents. This evidence emphasizes the importance of identifying novel risk factors and targets to optimize secondary preventive strategies. Lipoprotein(a) (Lp(a)) has emerged as an independent predictor of adverse events after ACS. New molecules such as anti-PCSK9 monoclonal antibodies, small interfering RNAs, and antisense oligonucleotides can reduce plasma Lp(a) levels and are associated with a long-term outcome benefit after the index event. The inflammatory stimulus and the inflammasome, pivotal elements in the development and progression of atherosclerosis, have been widely investigated in patients with coronary artery disease. More recently, randomized clinical trials including post-ACS patients treated with colchicine and monoclonal antibodies targeting cytokines yielded promising results in the reduction in major cardiovascular events after an ACS. Gut dysbiosis has also raised great interest for its potential pathophysiological role in cardiovascular disease. This evidence, albeit preliminary and needing confirmation by larger population-based studies, suggests the possibility of targeting the gut microbiome in particularly high-risk populations. The risk of recurrent ischemic events after ACS is related to the complex interaction between intrinsic predisposing factors and environmental triggers. The identification of novel risk factors and targets is fundamental to customizing patient clinical management with a precision medicine perspective.

## 1. Introduction

Coronary artery disease (CAD) is the leading cause of death worldwide, accounting for about 20% of cases in Europe [1]. In the last years, advances in preventive measures, pharmacological treatments [2,3,4], and percutaneous coronary interventional strategies [5,6] significantly improved life quality and expectancy after index myocardial infarction (MI), thus increasing the population of stable post-MI patients [7]. This novel epidemiological scenario emphasized the importance of implementing secondary preventive strategies, as these patients have a very high risk of further atherothrombotic events and mortality [8,9,10,11,12].

The control of traditional risk factors, the promotion of a healthy lifestyle, and antithrombotic therapy have been the cornerstones of secondary prevention after MI for decades. However, many patients continue to experience recurrent ischemic events despite lower residual cardiovascular risk and guideline-directed antithrombotic therapy [13].

In this new era of precision medicine, controlling traditional risk factors may not be sufficient for further reducing individual patient risk. Substantial attention has been focused on the identification of new risk factors [14,15,16] and new targets to develop personalized preventive strategies after an acute coronary syndrome (ACS). The aim of this review is to summarize the potential pathophysiological mechanisms underlying residual cardiovascular risk in post-ACS patients, describe novel risk factors associated with recurrent atherothrombotic events, and describe novel potential therapeutic targets that might be considered in this very high-risk clinical setting.

### 1.1. Risk of Recurrent Ischemic Events in Post-ACS Patients

The main pathogenetic mechanism of ACS is the rupture of thin-capped atherosclerotic plaque with a large lipid or necrotic core, with consequent exposure to prothrombotic material, a release of pro-thrombotic mediators, and ultimately, formation of a flow-limiting thrombus [17]. In the last years, we have understood that this mechanism is only a part of a much more complex pathophysiological framework. Intravascular imaging studies showed that subclinical plaque ruptures often occur in patients with CAD [18,19] and that some “vulnerable plaques” may evolve into a condition of stability, probably through the cycles of rupture and healing [20,21]. Indeed, many stable plaques causing significant flow-limiting stenoses show morphological features of previous plaque rupture and healing [22,23]. For reasons that we do not fully understand, in some patients, prothrombotic activation prevails over the healing response, leading to the formation of a flow-limiting thrombus and the clinical manifestation of an ACS. This pathophysiological evidence suggests that patients with a history of ACS represent a vulnerable niche of patients with CAD characterized by a biological predisposition to the occurrence of acute atherothrombotic events. 

Prior studies showed that post-ACS patients have a higher risk of ischemic events compared to stable CAD patients with no history of ACS [24,25]. The risk of recurrent events is higher during the first year after the index event but continues to increase over later years [9,26]. Data from large-scale real-world registries reported long-term annual rates of recurrent events between 4.4% and 6.7% in stable post-ACS patients [9,27]. All this evidence confirms that the “residual cardiovascular risk” remains clinically meaningful in post-ACS patients despite the adoption of optimal pharmacological therapy.

### 1.2. Residual Cardiovascular Risk after ACS: Limitations of Traditional Risk Factors

The estimate of individual residual risk after ACS is a complex multiparametric process that should theoretically account for many aspects including lifestyle habits, thrombotic cascade, lipoprotein particles, and inflammatory cells. Traditionally, the thrombotic and low-density lipoprotein cholesterol (LDL-C) pathophysiological pathways have been identified as the leading mechanisms involved in ACS recurrence, and many drugs have been developed for controlling them. The control of thrombotic phenomena after ACS patients refers to a holistic prothrombotic milieu that goes beyond the simple prevention of stent thrombosis, encompassing instead the risk of the destabilization of non-culprit lesions [28]. Dual antiplatelet therapy (DAPT) is the cornerstone of medical therapy after ACS. To overcome “residual thrombotic risk”, long-term DAPT strategies have also been developed for selected very-high-risk patient cohorts [29]. In the PEGASUS-TIMI 54 (Prevention of Cardiovascular Events in Patients with Prior Heart Attack Using Ticagrelor Compared to Placebo on a Background of Aspirin–Thrombolysis in Myocardial Infarction 54) trial, long-term DAPT with ticagrelor in addition to aspirin provided a 16% reduction in adverse ischemic events in post-MI patients [29]. The benefits of a prolonged DAPT were particularly relevant in patients with high atherosclerotic burden, such as those with coexistent peripheral artery disease [30]. Furthermore, the benefit of prolonged DAPT in PEGASUS was observed regardless of prior coronary stenting [31]. The reduction in ischemic events was counterbalanced by an increase in major bleeding, albeit with no difference in terms of fatal and intracranial bleeding.

An alternative strategy for the long-term prevention of recurrent thrombotic events is the combination of antiplatelet and low-dose anticoagulant agents. In the COMPASS (Cardiovascular Outcomes for People Using Anticoagulation Strategies) trial, the use of low-dose rivaroxaban (2.5 mg twice daily) in addition to aspirin in patients with atherosclerotic disease (including 61.8% of post-MI patients) significantly reduced the incidence of ischemic events at follow-up at the cost of higher rates of major bleeding [32].

These results emphasize the importance of a personalized approach based on the assessment of net ischemic and bleeding risks instead of a “one-size fits all” strategy. This may be difficult due to the substantial overlap of the predictors of ischemic and bleeding events and the difficulties inherent in balancing their individual contributions in each patient [33,34,35].

LDL-C has been recognized as a causal factor for atherosclerotic disease [36]. Landmark trials showed that LDL-c-lowering agents, such as statins and ezetimibe, significantly reduced the incidence of adverse ischemic events in patients with CAD [37,38]. However, several studies on lipid-lowering therapies showed a persistent residual risk of ischemic events despite an aggressive LDL-C control [37,39,40]. The issue of “residual cholesterol risk” has been partially addressed with the introduction of proprotein convertase subtilisin/kexin type 9 (PCSK9) inhibitors that dramatically reduce LDL-c levels when added to statin therapy, showing a significant reduction in cardiovascular events in patients with CAD [41,42]. However, a subanalysis from the FOURIER (Further Cardiovascular Outcomes Research with PCSK9 Inhibition in Subjects with Elevated Risk) trial showed that in patients treated with PCSK9 inhibitors, although an aggressive reduction in LDL-c up to 20–30 mg/dL was achieved, the high-sensitivity C-reactive protein (hsCRP) remained a strong predictor of recurrent ischemic events, confirming the central role of inflammation in atherogenesis in spite of optimal LDL-c levels [13,43].

These data demonstrated that we are still missing some pieces of the complex puzzle of residual cardiovascular risks. We will try to discuss the role of emerging risk factors of recurrent atherothrombotic events in post-ACS patients below, with a focus on potential therapeutic strategies.

## 2. Lipoprotein(a)

Lipoprotein(a) (Lp(a)) is a plasma lipoprotein composed of an LDL-rich particle with an apolipoprotein B100 molecule bound via a single disulphide bond to apolipoprotein(a) (apo(a)), a plasminogen-like glycoprotein. This is a pathogenetic component of Lp(a) containing, similarly to plasminogen, several three-dimensional domains called kringles [44]. 

### 2.1. Genetics and Activity of Lp(a)

Lp(a) concentrations vary from <0.1 mg/dL to >300 mg/dL (<0.2 to 750 nmol/L) and are mostly (>90%) determined by genetic variability in the *LPA* locus with negligible influence attributable to dietary and environmental factors [44]. Kringle-IV (K-IV) polymorphisms are responsible for the great variability in the plasma concentration of Lp(a). A low number (<23) of K-IV repeat polymorphisms are associated with smaller apo(a) isoforms and higher plasma Lp(a) levels than a higher number of K-IV repeats and larger apo(a) isoforms [45].

In vitro and animal studies have emphasized the central role of Lp(a) in different pathophysiological pathways of atherosclerosis, including foam cell formation, smooth muscle cell proliferation, inflammation, and plaque instability [46]. Indeed, Lp(a) not only appears to contribute to atherosclerosis through the same mechanisms of LDL but also presents additional apo(a)-related effects. Apo(a), due to its close homology with plasminogen, competitively inhibits fibrinolysis by binding fibrin and thus promotes thrombotic phenomena [47]. Moreover, apo(a) contains lysine binding sites, permitting its interaction with the exposed portions of the damaged endothelium and its penetration and accumulation in subintimal spaces and conferring a particular propensity of Lp(a) for vascular walls [48]. Furthermore, Lp(a) is one of the main carriers of oxidized phospholipids (Ox-PL) within the vessel wall, enhancing atherogenesis and inflammation [49]. In an in vivo study, patients with elevated Lp(a) showed an increased accumulation of 18-fluorodeoxyglucose in the carotid arteries and aorta, indicating the inflammation of arterial walls. In addition, the monocytes of these patients showed increased responsiveness in the production of proinflammatory cytokines and a marked tendency to transmigrate, which could reflect their capacity to penetrate the vessel wall [50]. Moreover, Lp(a) seems to upregulate the expression of proinflammatory genes and promote the release of inflammatory cytokines and chemokines, such as interleukin-8 and monocyte chemoattractant protein-1 [51,52].

### 2.2. Lipoprotein(a) as a Cardiovascular Risk Factor

Over the past two decades, many studies conducted in populations without a history of atherosclerotic cardiovascular disease (ASCVD), including observational studies, meta-analyses, mendelian randomization studies, and genome-wide association studies, have shown that high levels of Lp(a) are linearly associated with an increased risk of developing ASCVD [53,54,55,56,57]. The Copenhagen City Heart Study, which included 9330 subjects followed for 10 years, showed a linear increase in the risk of cardiovascular events as the level of Lp(a) increased, reaching a 3.6-fold greater risk in subjects with extremely elevated Lp(a) levels (>120 mg/dL) [53]. A recent Mendelian randomization study matching 9015 patients with acute MI and 8629 controls from the Pakistan Risk of Myocardial Infarction Study (PROMIS) suggested that both smaller apo(a) isoform size and increased Lp(a) concentration are independent and causal risk factors for CAD [58]. An individual patient data meta-analysis, which included 29,609 patients from seven randomized clinical trials, showed that high Lp(a) levels are linearly associated with the risk of cardiovascular adverse events independently of LDL levels and statin treatment [59]. This evidence established Lp(a) as a risk factor and a potential therapeutic target for primary prevention in healthy subjects. Indeed, the European Society of Cardiology (ESC) and European Atherosclerosis Society (EAS) recommend measuring Lp(a) at least once in adults, preferably in the first lipid profile, and they suggest a desirable plasma level of <30 mg/dL (IIaC) [60].

### 2.3. Usefulness of Lipoprotein(a) in Secondary Prevention

In patients with established CAD, the prognostic role of Lp(a) levels has long been debated. Recently, several studies show the association between Lp(a) and adverse events in ASCVD settings [8,42,61,62,63,64,65,66,67,68]. A meta-analysis that included three randomized trials and eight secondary prevention studies, including a total of 18,978 patients, found that subjects with Lp(a) levels above the highest quintile were at increased risk of cardiovascular events but with a high heterogeneity between studies. However, this correlation was not significant for subjects whose plasma LDL levels were <130 mg/dL [65].

In the Copenhagen General Population Study, the 5-year risk for recurrent major adverse cardiovascular events (MACEs) in subjects with pre-existing ASCVD was higher for Lp(a) levels ≥50 mg/dL, independently of LDL plasma levels [69]. Recently, a pre-specified analysis from the ODISSEY Outcomes trial, which randomized 18,924 post-ACS patients on high-intensity statin treatment relative to alirocumab vs. placebo, found that baseline Lp(a) levels were linearly associated with MACE (composite of coronary death, MI, stroke, or unstable angina), fatal and non-fatal MI, and cardiovascular death, independently of baseline LDL values. They also found that the reduction in Lp(a) levels given by alirocumab independently reduced the risk of MACE [66]. Furthermore, a recent observational study including 12,064 consecutive unselected real-world patients who underwent PCI with drug-eluting stents showed that high plasma levels of Lp(a) were significantly and independently associated with a higher risk of recurrent ischemic events and repeated revascularization at a long-term follow-up [67]. More recently, Lp(a) emerged as an independent predictor of recurrent MI after ACS, particularly in patients without diabetes [8,64]. 

These pieces of evidence emphasize the importance of including baseline Lp(a) plasma levels in the prognostic stratification of patients with established ASCVD and the increasing need for pharmacological agents to target this molecule.

### 2.4. Current and Emerging Therapies for High Lipoprotein(a)

To date, the only available and effective therapy for reducing Lp(a) levels is lipoprotein apheresis, indicated for secondary prevention in patients with extremely high Lp(a) levels and recurrent cardiovascular events despite optimal medical therapy. Since lipoprotein apheresis is an invasive, expensive, and not risk-free procedure, it is used only in a few selected patients in third-level dedicated centers [70].

There are no approved pharmacological agents for Lp(a) nor preliminary data from randomized clinical trials on Lp(a) reduction. Although statins are a milestone treatment of dyslipidaemia, they have failed to show any effect in lowering Lp(a) plasma concentrations [71]. In the AIM-HIGH (Atherothrombosis Intervention in Metabolic Syndrome with Low HDL/High Triglyceride and Impact on Global Health Outcomes) trial, niacin reduced Lp(a) plasma levels by 21% but without a significant influence on the cardiovascular risk [62].

Mipomersen, an antisense oligonucleotide (ASO) targeting apolipoprotein B synthesis, has been shown to reduce Lp(a) levels by 26% in four phase 3 trials, but its clinical impact remains unclear [72].

PCSK9 inhibitors are currently the most promising molecules liable to reduce the plasma levels of Lp(a). A post hoc analysis from the FOURIER trial, in which 25,096 patients with stable CAD were randomized relative to evolocumab vs. placebo, showed a reduction in Lp(a) plasma levels of 27% with evolocumab associated with a significant decrease in adverse cardiovascular outcomes (composite of coronary heart disease, death, myocardial infarction, and urgent revascularization) with an absolute risk reduction of 2.49% and number needed to treat of 40 up to 3 years in patients with Lp(a) plasma values above the median 37 nmol/L serum level [68]. Similarly, a post hoc analysis of the ODISSEY Outcomes trial showed that alirocumab was associated with a reduction in Lp(a) plasma levels of 23%, leading to a significant reduction in long-term cardiovascular adverse events in patients with ACS; however, this outcome benefit was principally due to the reduction in non-fatal MI, ischemia-driven coronary revascularization, and major PAD events rather than mortality. Moreover, in this study, baseline Lp(a) levels were independently associated not only with the patients’ cardiovascular risk but also with the expected benefit from alirocumab treatment [73].

The ORION-10 and ORION-11 trials showed that Inclisiran, a small-interfering RNA (siRNA) PCSK9 inhibitor, was associated with a 20% reduction in Lp(a) levels, but the trial did not investigate the effect on cardiovascular outcomes [74]. To date, PCSK9 inhibitors are recommended by ESC/EAS guidelines in subjects with familial hypercholesterolemia and high Lp(a) (IIa) [75].

Eventually, newer emerging therapies targeting specifically apo(a) via ASOs (Pelacarsen) and siRNAs (Olpasiran) are being investigated in phase II and III trials, with encouraging preliminary results in terms of safety and efficacy in reducing plasma Lp(a) levels, and data on the clinical outcome are expected in the coming years [76,77,78].

Table 1 summarizes the emerging pharmacological therapies for lowering Lp(a) plasma levels.

## 3. Inflammation

Atherosclerosis is an unsolved progressive chronic inflammatory process related to cholesterol deposits within the arterial intima, which plays a central role in developing type I MI [79]. The outcome benefit associated with anti-inflammatory therapies in patients with established ASCVD emphasizes the importance of inflammation in the development and progression of atherosclerosis independently of the optimal control of other traditional cardiovascular risk factors, including LDL cholesterol [80,81,82,83].

### 3.1. Inflammation, Inflammasome, and Atherosclerosis

The damaged and activated endothelium in the initial phase of atherosclerosis favors the recruitment of circulating monocytes, T-cells, and mast cells that infiltrate the atheroma [84,85] and participates in the release of a high quantity of inflammatory cytokines [86,87]. In this environment, there is a vicious cycle related to apoptised intimal cells, which are not engulfed and efficiently cleared by macrophages and favor the build-up of the necrotic core. The persistence of these necrotic elements further promotes the activation and release of more inflammatory molecules by the cells of innate immunity [88,89]. Therefore, inflammation plays a crucial role in the pathogenesis and progression of atherosclerosis and actively participates in increasing the residual risk in patients with history of ACS. 

A meta-analysis involving 160,309 subjects from 54 long-term prospective studies showed a continuous association between CRP levels and CAD (RR 1.32; 95% CI 1.18 to 1.49) [90]. Moreover, a secondary analysis from the CANTOS (The Canakinumab Anti-Inflammatory Thrombosis Outcomes Study) trial, which evaluated the efficacy of canakinumab vs. placebo in 1061 patients with a history of MI, showed that patients who reached high-sensitivity CRP levels <2 mg/dL had a significant reduction (25%) in MACE (HR: 0.75, 95% CI 0.66 to 0.85) than those who had high-sensitivity CRP plasma levels >2 mg/dL [91].

In recent years, the inflammasome, which indicates the mechanism for caspase-1 activation and interleukin-1b processing, emerged as one of the main actors and facilitators of the inflammatory process [92,93]. In particular, the nucleotide-binding oligomerization domain and leucine-rich repeat-containing receptor (NLR) family pyrin domain-containing 3 (NLRP3) inflammasome have emerged to be big players in the development of atherosclerosis [93].

### 3.2. NLRP3 in Atherosclerosis and Cardiovascular Risk

Different stimuli can activate NLRP3 [94]. In the peculiar context of atherosclerosis, shear stress on the endothelial wall, cholesterol crystals, and Ox-PL seem to be the primers of NLRP3 activation [95,96,97,98]. After activation, NLRP3 enhances via the caspase-1 activation of proinflammatory cytokine interleukin-1β (IL-1β), which is the circulating form of interleukin-1 (IL-1) cytokine [99]. IL-1 exhibits a wide variety of cardiovascular effects: 99 It induces the production of additional proinflammatory mediators, such as tumor necrosis factors, interleukin-6 (IL-6), and chemoattractant molecules, implicated in the tissue invasion of inflammatory cells into the atheroma. IL-6 drives the secretion of acute-phase reactants, such as fibrinogen and plasminogen activator inhibitor-1 (PAI-1), facilitating prothrombotic and antifibrinolytic impairment [100,101]. Notably, IL-1 facilitates its own gene expression, maintaining a self-sustained inflammatory response [102].

In experimental studies, IL-1 worsened ischemic-reperfusion injury after MI and enhanced negative cardiac remodeling; conversely, IL-1 inhibition mitigated negative cardiac remodeling and decreased acute phase protein release [103,104,105,106]. This evidence highlights the contribution of the NLRP3-mediated inflammatory cascade in the development and progression of atherosclerosis in patients with established ASCVD, thus increasing their residual risk of recurrent ischemic events.

### 3.3. Anti-Inflammatory Therapies in ASCVD Patients

Anti-inflammatory therapies have been proposed as potential game changers in the future management of patients with established ASCVD, and NLRP3 inflammasome and its products could be targets for future drugs. 

Among the anti-inflammatory agents, colchicine has been widely studied in patients with established ASCVD [82,83,107]. By inhibiting polymerization and microtubule formation, colchicine could interfere with NLRP3 formation [108]. Data from two large randomized controlled trials, COLCOT and LoDoCo2, showed that the daily 0.5 mg of colchicine significantly reduced the risk of adverse cardiovascular events by 23% and 30%, respectively, in cohorts of CAD patients [82,83]. Recently, a meta-analysis including 12,869 patients from 11 randomized controlled trials reported that the use of colchicine in patients with CAD was effective in reducing major adverse cardiovascular and cerebrovascular events (with an acceptable risk profile in terms of drug-related adverse events), but it did not significantly reduce the risk of cardiovascular and non-cardiovascular death. Moreover, a prior meta-analysis showed a higher, but not significant, incidence of non-cardiovascular deaths in patients treated with low-dose colchicine [109]. The 2021 ESC guidelines on cardiovascular disease prevention suggest that low-dose colchicine (0.5 mg daily) may be considered in secondary prevention, especially if the other risk factors are insufficiently controlled or if recurrent cardiovascular events occur under optimal therapy (IIb) [110]. 

Canakinumab, a human monoclonal antibody that can hamper the interaction between IL-1β and its receptor, has been tested in the CANTOS trial, which randomized 10,061 patients with established CAD to different doses of subcutaneous canakinumab (50 mg, 150 mg, and 300 mg every 3 months) vs. the placebo. The 150 mg dose significantly reduced the risk of adverse cardiovascular events vs. the placebo. However, the effect on the primary endpoint (composite of non-fatal MI, non-fatal stroke, and cardiovascular death) was largely driven by a reduction in non-fatal MI, whereas a non-significant reduction in long-term mortality was observed [81]. Therefore, these results should be interpreted with caution since non-fatal MI is not validated as a valid surrogate of all-cause or cardiovascular mortality in clinical trials enrolling patients with CAD [111]. Moreover, a CANTOS pre-specified analysis showed that canakinumab led to a significant reduction in IL-6 and hs-CRP [91]. However, caution is necessary since canakinumab may alter immune homeostasis, as it acts systemically; in fact, patients treated with this drug reported a higher incidence of infections, even fatal ones, compared to the placebo [81].

Methotrexate (MTX), a competitive folic acid antagonist, is a widely used drug in rheumatic diseases. A meta-analysis including 10 cohort studies conducted on patients with rheumatoid arthritis receiving MTX showed a significant reduction in cardiovascular events [112]. However, the recent Cardiovascular Inflammation Reduction trial (CIRT), conducted in 4786 patients with stable atherosclerosis receiving low doses of MTX (15–20 mg), did not show a significant reduction in cardiovascular events, including hs-CRP, IL-1β, and IL-6 plasma levels [113]. MTX delivered in LDL nanoparticles is currently being studied in ASCVD patients in two ongoing trials, including stable CAD patients and STEMI patients (NCT04616872; NCT 03516903). 

Anakinra, an IL-1 receptor antagonist, significantly reduced the acute inflammatory response, as demonstrated in a phase II trial including 99 patients with STEMI and acute decompensated heart failure; however, no significant differences were found in terms of clinical outcomes compared to the placebo [114].

Tocilizumab and Ziltivekimab, two human IgG1 monoclonal antibodies targeting IL-6 and the IL-6 ligand, have recently been studied in two different randomized controlled trials [115,116]. In the ASSAIL-MI (ASSessing the effect of Anti-IL-6 treatment in MI) trial, Tocilizumab at a 280 mg dose increased myocardial salvage in STEMI patients [115]. In the Trial to Evaluate Reduction in Inflammation in Patients with Advanced Chronic Renal Disease Utilizing Antibody Mediated IL-6 Inhibition (RESCUE), Ziltivekimab, a monoclonal antibody inhibiting the IL-6 ligand, markedly reduced the serum biomarkers of inflammation and thrombosis in patients with advanced chronic kidney disease [116].

Arglabin is a molecule isolated from different plant species that seems to interfere with the stimulation of NLRP3 in macrophages, and in animal models, it has been shown to significantly reduce NLRP3-related inflammatory molecules and has a beneficial impact on lipid profiles [117]. MCC950 is a selective NLRP3 inhibitor that, in several animal models, has been shown to decrease IL-1β production, reduce atherosclerotic plaque size, and reduce the infarct area in experimental models of myocardial infarction [80,118]. 

Although anti-inflammatory therapies could represent the way forward for the treatment of patients with established ASCVD, their long-term efficacy and safety is still unknown, and some of them are forbiddingly expensive. In the forthcoming years, it will be necessary to find drugs that are both safe and effective up until the long-term follow-up and with affordable costs.

Anti-inflammatory and immune therapies tested in patients with established CAD are summarized in Table 2.

## 4. Gut Microbiota

The gut microbiota consists of innumerable microbes that colonize the entire digestive tract from the stomach down to the colon. Its composition could be influenced by genetic, dietary, and environmental factors [119,120]. The bacteria and viruses that make it up cause a continuous inflammatory trigger, which is outweighed by the host’s innate immunity [121]. Gut dysbiosis, a condition in which the microbial composition of microbiota is altered, has been linked to numerous pathological conditions as well as intestinal diseases, type 2 diabetes, obesity, and ischemic stroke [122]. Recently, emerging interest has grown on the possibility that the permeation of specific bacterial products of altered microbiota, such as lipopolysaccharide (LPS) and trimethylamine-N-oxide (TMAO), within the systemic circulation may promote the onset of atherothrombotic and ischemic cardiovascular events [123]. Animal models with *ApoE^−/−^* mice showed that increased plasma levels of TMAO lead to an augmented atherosclerotic burden [124]; in addition, a reduction in TMAO plasma levels showed an abortion of the atherosclerotic effect with a decrease in plaque size [125]. However, clinical studies in humans showed conflicting results upon the correlation between TMAO, atherosclerotic burden, and cardiovascular events; thus, further studies are needed to clarify the role of TMAO in the development and progression of atherosclerosis [126,127,128,129,130]. A nested case–control study from the PEGASUS-TIMI 54 trial including 597 post-MI patients and 1206 controls showed that higher TMAO levels were associated with cardiovascular death and stroke but not with recurrent MI in patients with a history of prior MI [131].

Emerging evidence suggests that low-grade endotoxemia driven by circulating LPS could lead to atherothrombosis and cardiovascular disease [129]. LPS is detected by the receptors of the immune system, such as toll-like receptors, a group of membrane receptors involved in atherosclerotic and thrombotic processes [132]. Previous studies linked the LPS circulating levels with a long-term risk of developing ASCVD; however, these studies were conducted in healthy subjects, and the role of LPS in patients with established ASCVD has not yet been evaluated [133,134,135].

A case–control study including 50 patients diagnosed with STEMI, 50 patients with stable CAD, and 50 healthy controls demonstrated an augmented gut permeability during myocardial ischemia and, consequently, a significant role of circulating LPS in favoring thrombus growth during STEMI [136]. Recently, a study conducted in a population of post-STEMI patients showed that the permeation of LPS in the systemic circulatory system during the acute phase of MI significantly correlated with future adverse cardiovascular outcomes at 3 years follow-up [137]. Despite these results, further efforts are needed to better understand the potential role of gut dysbiosis on the individual cardiovascular risk of patients with established ASCVD. 

The next challenge could be the possibility of directly targeting the gut microbiome in high-risk patients. Indeed, fibre-rich diets such as a Mediterranean diet, prebiotics, probiotics, and faecal microbiome transplantation have been shown to positively regulate gut permeability and can reduce circulating levels of bacterial products. However, this evidence is limited to animal models and small-sampled clinical studies [123,129].

## 5. Conclusions and Future Perspectives

Secondary prevention in post-ACS patients is a major health problem. Despite significant improvements in behavioral, pharmacological, and invasive treatments, these patients still show a high risk of recurrent atherothrombotic events and mortality. The residual risk of post-ACS patients is related to the interaction between intrinsic predisposing factors and environmental triggers, combined in a complex pathophysiological framework. The impossibility to simplify an articulated biological phenomenon calls for a tailored clinical approach for post-ACS patients, including the identification of novel risk factors and targets. 

Lp(a) is a non-traditional cardiovascular risk factor that has progressively assumed relevance over the last years, and ongoing trials will clarify the best use of Lp(a) as a therapeutic target after ACS. 

Chronic inflammation is recognized as a key pathogenetic element of atherosclerosis, and recent evidence from large-scale randomized trials supports the use of anti-inflammatory drugs for secondary prevention in patients with recurrent ACS. However, further studies are needed to definitely clarify the safety and cost-effectiveness profile of these therapeutic options. 

The characterization of the systemic implications of gut dysbiosis is a new field of translational medicine. The link between dysfunctional microbiota and atherothrombosis could provide the rationale for novel therapies in the future.

The mission of the cardiology community is to pursue the search for novel risk factors and targets, develop new therapies, and ultimately improve patient outcomes after ACS. 

## Figures and Tables

**Table 1 jcm-12-02161-t001:** Emerging therapies for lowering Lp(a) serum levels.

Drug	Principal Mechanism of Action	Mechanism of Lp(a) Lowering	Study	Design	Population	Findings
Evolocumab	Monoclonal antibody inhibiting LDL-R degradation by targeting PCSK9	Inhibition of apo(a) secretion	FOURIER (NCT01764633)	Sub-analysis of phase III RCT	25,096 stable-CAD pts	Evolocumab significantly reduced Lp(a) by a median of 26.9% [54]; Evolocumab reduced the risk of death, MI, or PCI by 23% in patients with a baseline Lp(a) > median value [54]
Alirocumab	Monoclonal antibody inhibiting LDL-R degradation by targeting PCSK9	Inhibition of apo(a) secretion	ODISSEY Outcomes (NCT01663402)	Sub-analysis of phase III RCT	18,924 post-ACS pts	Alirocumab significantly reduced Lp(a) by 23% [61]; Alirocumab independently reduced the risk of CV adverse outcomes [61]
Inclisiran	siRNA inhibiting LDL-R degradation targeting PCSK9	Inhibition of apo(a) secretion	ORION-10 ORION-11 (NCT03399370; NCT03400800)	Phase III RCT	2178 pts with ASCVD (1975) or ASCVD risk equivalent (203)	Inclisiran reduces Lp(a) plasma levels by 19–22% [62]; the effect on CV outcomes is unknown
Mipomersen	ASO inhibiting apo(B) synthesis	-----	Four phase III trials (NCT00607373; NCT00706849; NCT00770146; NCT00794664)	Four phase III RCTs	382 pts diagnosed with: - HoFH (51), - HeFH with CAD (123), - severe HC (57), - HC at risk of CAD (151)	Mipomersen reduced Lp(a) plasma levels from 20% to 40% [60]; the effects on CV outcomes are unknown
Olpasiran	siRNA targeting apo(a) mRNA and leads to degradation	-----	OCEAN[a]-DOSE (NCT04270760)	Phase II RCT	281 pts with high Lp(a) and ASCVD	Olpasiran reduced Lp(a) plasma levels from 67% to 97% [66]; the effect on CV outcomes is unclear
Pelacarsen	ASO targeting apo(a) mRNA and leads to degradation	-----	AKCEA-APO(a)-LRx (NCT03070782)	Phase II RCT	286 pts with high Lp(a) and CVD	Pelacarsen reduced Lp(a) plasma levels by 80% [65] A phase III RCT evaluating the effect on CV outcomes is ongoing (NCT04023552)

ACS, acute coronary syndromes; Apo(a), apolipoprotein(a); Apo(B), apolipoprotein(B); ASCVD, atherosclerotic cardiovascular disease; ASO, antisense oligonucleotide; CAD, coronary artery disease; CV, cardiovascular; CVD, cardiovascular disease; HC, hypercholesterolemia; HeFH, heterozygous familial hypercholesterolemia; HoFH, homozygous familial hypercholesterolemia; LDL-R, low-density lipoprotein receptor; Lp(a), lipoprotein(a); MI, myocardial infarction; PCI, percutaneous coronary intervention; PCSK9, proprotein convertase subtilisin/kexin type 9; pts, patients; RCT, randomized controlled trial.

**Table 2 jcm-12-02161-t002:** Anti-inflammatory and immune therapies in patients with established CAD.

Drug	Principal Mechanism of Action	Study	Design	Population	Findings
Colchicine	Inhibition of microtubules polymerisation	COLCOT (NCT02551094) LoDoCo2 (ACTRN12614000093684)	Phase III RCT	4745 pts with recent MI (COLCOT) 5522 pts with chronic CAD (LoDoCo2)	Colchicine 0.5 mg significantly reduced the risk of ischemic CV events [70,71]
Canakinumab	anti-IL-1β monoclonal antibody	CANTOS (NCT01327846)	Phase III RCT	10,061 pts with previous MI and hs-CRP >2 mg/L	Canakinumab 150 mg every 3 months significantly reduced the rate of recurrent CV events [69]
Methotrexate	Folate pathway antagonist	CIRT (NCT01594333)	Phase III RCT	4786 pts with previous MI or MVD	MTX 15–20 mg weekly did not reduce levels of IL-1β, IL-6, or CRP and did not reduce CV events [99]; a phase III RCT evaluating MTX delivered in LDL-nanoparticles is ongoing (NCT04616872)
Anakinra	Inhibition of the interaction between IL-1 and IL-1R	VCU-ART3 (NCT01950299)	Phase II RCT	99 pts with STEMI and acute HF	Anakinra significantly reduces the systemic inflammatory response after STEMI; no difference in terms of CV outcomes [100]
Tocilizumab	Inhibits IL-6R	ASSAIL-MI (NCT03004703)	Phase II RCT	199 pts with STEMI	Tocilizumab increased myocardial salvage; no difference in terms of CV outcomes [101]
Ziltivekimab	Inhibits IL-6 ligand	RESCUE (NCT03926117)	Phase II RCT	264 pts with CKD	Ziltivekimab reduced biomarkers of inflammation and thrombosis [102]; phase III RCT evaluating the effect on CV outcomes is ongoing (NCT05021835)

CAD, coronary artery disease; CKD, chronic kidney disease; CRP, C-reactive protein; CV, cardiovascular events; HF, heart failure; hs-CRP, high-sensitivity C-reactive protein; IL-1, interleukin-1; IL-1R, interleukin-1 receptor; IL-1β, interleukin-1β; IL-6, interleukin-6; IL-6R, interleukin-6 receptor; LDL, low-density lipoprotein; MI, myocardial infarction; MTX, methotrexate; MVD, multivessel disease; STEMI, ST-elevation myocardial infarction; pts, patients; RCT, randomized controlled trials.

## Data Availability

Not applicable.

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
