# Peer review of "Secondary Cardiovascular Prevention after Acute Coronary Syndrome: Emerging Risk Factors and Novel Therapeutic Targets"

_jcm, 2023, doi:10.3390/jcm12062161_

Round 1

Reviewer 1 Report

The review paper from Silverio et al. brings important insights about emerging risk factors and potential novel therapies for secondary prevention after ACS. The paper is well written, and the topic is of high relevance to the clinical practice, not only to the Cardiology community, but also to the general audience. There are some suggestions which may improve the paper further, as outlined below:

1)    In page 2, line 62, the paragraph starts with “We have learned…” I have no objection for using first person in original articles, but it seems odd for a review article. Please, consider change.

2)    In page 3, line 100, authors may add also that “Furthermore, the benefit of prolonged DAPT in PEGASUS was observed regardless of prior coronary stenting” (Furtado RHM, et al. Eur Heart J 2020 ;41(17):1625-1632)

3)    In the paragraph about COMPASS trial, I suggest to add which dose is “low-dose rivaroxaban”. Although people used to the results of this trial know that, the general audience may not be aware of the so-called “COMPASS-dose” of 2.5 mg BID.

4)    In page 4, lines 181-182, I suggest to add what is the class of recommendation and level of evidence for measuring Lp(a) by the ESC guidelines.

5)    Page 5, line 197, please, add between parentheses after MACE: “(i.e., the composite of coronary death, MI, stroke or unstable angina)”. Some readers may not know the primary endpoint of the ODYSSEY OUTCOMES trial.

6)    Page 7, lines 302-304. I suggest to add that, in this meta-analysis, colchicine was not associated with a reduction in CV death or all-cause death, and that, in another prior meta-analysis (Fiolet ATL et al. Eur Heart J 2021; 42(28):2765-2775), there was a signal for potential increase in non-CV deaths with colchicine.

7)    In the section about gut microbiota, authors could cite the paper from Bencer G et al. (J Am Heart Assoc 2020 May 18;9(10):e015331) suggesting a possible association between higher levels of TMAO and CV events among patients with history of MI.

Minor points

1)    Page 2, line 54: Replace “may be not” for “may not be”

2)    Page 2, line 87: Should be “account for may aspects…”

3)    Page 4, line 184: Suggest to add “reduction” after lipoprotein(a)

4)    Page 7, line 326: There are two ongoing trials with MTX-nanoparticle indeed: one in chronic ASCVD (NCT 04616872) and another one in acute STEMI (NCT 03516903).

Reviewer 2 Report

This is a nice review of emerging risk factors and novel therapeutic targets for secondary prevention. There is significant interest in the CV community in reducing residual risk, which has thus far been elusive. I would like to see the authors provide more context on the endpoints included in the trials and the absolute risk reduction (NNT) for these novel therapeutics. In every case presented they provide relative risk ratios to suggest possible benefits of therapy; however, this review would look much more sobering (which is the actual reality) if they provided some context on NNT in these studies and what components of the endpoints were actually reduced. For example, none of the trials on emerging therapeutics show significant mortality reduction signals and the significance of nonfatal MI as an endpoint has been recently questioned as it no longer associates with mortality to a significant extent (David Brown, et al's paper in JAMA IM). The inclusion of revascularization in many of these composite endpoints is very soft. This manuscript would be much stronger if the authors would comment on these concerns and they should do it in each section where trial data on novel therapeutics is presented. If this information was provided, in my opinion, it would make this a worthwhile contribution to the literature.

Round 2

Reviewer 2 Report

The authors have provided suitable revisions to satisfy my concerns and I feel the manuscript has been strengthened considerably.